# Prognostic and Predictive Utility of GPD1L in Human Hepatocellular Carcinoma

**DOI:** 10.3390/ijms241713113

**Published:** 2023-08-23

**Authors:** Philip K. H. Leung, Bibek Das, Xiaoyu Cheng, Munir Tarazi

**Affiliations:** Division of Surgery, Department of Surgery and Cancer, Faculty of Medicine, Imperial College London, Hammersmith Hospital, Du Cane Road, London W12 0NN, UK; p.leung@imperial.ac.uk (P.K.H.L.); bibek.das07@imperial.ac.uk (B.D.); xiaoyu.cheng22@imperial.ac.uk (X.C.)

**Keywords:** hepatocellular carcinoma, glycerol-3-phosphate dehydrogenase 1-like, prognosis, tumorigenesis, metabolic dysregulation

## Abstract

Hepatocellular carcinoma (HCC) is a major cause of cancer-related deaths worldwide. GPD1L, a member of the glycerol-3-phosphate dehydrogenase family, has emerged as a potential tumour suppressor gene, with high expression associated with a favourable prognosis in various cancers. Despite an intriguing inverse relationship observed with HCC, the precise role and underlying function of GPD1L in HCC remain poorly understood. Here, we aimed to investigate the prognostic significance, molecular characteristics, and predictive potential of GPD1L overexpression in HCC. Analysis of independent datasets revealed a significant correlation between high GPD1L expression and poor survival in HCC patients. Spatial and single cell transcriptome datasets confirmed elevated GDP1L expression in tumour tissue compared to adjacent normal tissue. GPD1L exhibited increased expression and promoter demethylation with advancing tumour stage, confirming positive selection during tumorigeneses. GPD1L overexpression was associated with metabolic dysregulation and enrichment of gene sets related to cell cycle control, epithelial-mesenchymal transition, and E2F targets. Moreover, we demonstrated an inverse correlation between GPD1L expression and therapeutic response for three therapeutic agents (PF-562271, Linsitinib, and BMS-754807), highlighting its potential as a predictive biomarker for HCC treatment outcomes. These data provide insights into the prognostic significance, molecular characteristics, and predictive potential of GPD1L in HCC.

## 1. Introduction

Hepatocellular carcinoma (HCC) is a prevalent form of liver cancer with a significant impact on global health. It is the sixth most common cancer and the third leading cause of cancer-related deaths worldwide [1]. Despite advances in early detection and treatment, the prognosis for HCC patients remains poor with a 5-year survival rate of 17%, primarily due to late-stage diagnosis and limited treatment options [2]. Therefore, there is an urgent need to identify novel therapeutic targets and develop more effective treatment strategies. Understanding the molecular mechanisms underlying HCC development and progression is crucial for improving therapeutic strategies and patient outcomes.

Glycerol-3-phosphate dehydrogenase 1-like (GPD1L) is a member of the glycerol-3-phosphate dehydrogenase family, which plays a critical role in cellular energy metabolism and redox homeostasis [3]. GPD1L has emerged as a significant gene in HCC, where previous studies have identified that high GPD1L expression is associated with poor prognosis in HCC [4,5]. Interestingly, GPD1L acts as a tumour suppressor in many other cancers, where high expression levels are associated with a favourable prognosis [6,7,8,9,10]. The inverse relationship between high GPD1L expression and poor prognosis in HCC, contrasting with other cancers where high expression is associated with a favourable prognosis, highlights the need for further investigation into the underlying molecular mechanisms.

One of the key mechanisms is the regulation of redox homeostasis, whereby GPD1L counteracts oxidative stress and maintains cellular redox balance [11]. This activity is particularly important in the context of mitochondrial stress, as mitochondrial dysfunction is frequently observed in cancer cells [12]. The protective effect of GPD1L against redox stress, particularly mitochondrial stress, suggests its potential in adapting to the high metabolic demands observed in certain tumours. Furthermore, GPD1L has been identified as a target of miR-210, a key regulator under hypoxic conditions [13]. Elevated miR-210 levels resulted in reduced GPD1L expression, leading to the stabilisation of hypoxia-inducible factor 1 alpha (HIF-1α) and the subsequent activation of hypoxia-responsive pathways [14]. This regulatory axis suggests that miR-210-mediated downregulation of GPD1L is a critical event in the adaptive response to hypoxia, highlighting the intricate interplay between GPD1L, miR-210, and hypoxia signalling pathways.

The aim of this study is to investigate the prognostic significance and molecular characteristics of GPD1L in HCC, including its correlation with clinical outcomes, its role in tumorigenesis and progression, and its potential as a predictive biomarker for treatment response. Differential gene expression analysis, survival analysis, and functional enrichment analysis were performed to investigate the role of GPD1L in HCC. Drug response prediction was also conducted using HCC cell lines in silico and in vitro.

## 2. Results

### 2.1. GPD1L Gene Expression as a Prognostic Biomarker in HCC

We initially investigated the association between the tumour GPD1L gene expression and clinical outcomes in HCC. Analysis of the TCGA-LIHC primary tumour dataset (n = 370) revealed a significant correlation between high GPD1L mRNA expression and several clinical factors associated with poor prognosis, including younger age, female gender, advanced stage, positive resection margins, and worse performance status (Table 1 and Appendix A). Notably, no significant differences were observed in the primary risk factors for HCC or synthetic liver function (INR). These findings are consistent with results from three additional datasets (HCCDB v2.0) that included matched transcriptomic and survival data, demonstrating that high GPD1L mRNA expression is associated with worse overall survival (OS) (log rank *p* < 0.05) (Figure 1).

Furthermore, we employed a Cox proportional hazards model to assess the prognostic significance of GPD1L mRNA expression in the TCGA dataset. The results demonstrated that high GPD1L mRNA expression was a significant poor prognostic factor for OS in both univariate (hazard ratio [HR] 1.72; confidence interval [CI] 1.21–2.44; *p* = 0.002) and multivariate analysis (HR 1.60; CI 1.02–2.52; *p* = 0.042) (Appendix A). Regarding the progression-free interval (PFI), high GPD1L mRNA expression showed poor prognostic significance in univariate (HR 1.35; CI 1.01–1.82; *p* = 0.043), but not in multivariate analysis (HR 1.06; CI 0.74–1.52; *p* = 0.7) (Appendix A).

In addition, a strong significant association between high GPD1L mRNA and adverse histological characteristics such as vascular invasion and higher histological grade was found in the TCGA-LIHC dataset (Appendix A). The association between high GPD1L expression and vascular invasion was corroborated in an independent dataset (GSE19977) (Appendix A). No significant difference in GPD1L expression was found with N+ and M+ disease in the TCGA-LIHC dataset due to low numbers. 

Overall, these consistent findings across independent datasets indicate that GPD1L gene expression is prognostic in HCC and support existing evidence that GPD1L expression is a marker of aggressive tumours.

### 2.2. Positive Selection for GPD1L Overexpression in HCC

Next, we conducted a pan-cancer analysis using TCGA data to investigate GPD1L expression patterns. GPD1L exhibited downregulation in all tumour samples compared to adjacent normal tissue, except for hepato-biliary tumours (HCC and cholangiocarcinoma) (Figure 2A and Appendix A). Specifically focusing on HCC, GPD1L mRNA levels were found to be increased in tumour tissues relative to adjacent normal tissue in 13 out of 18 HCC datasets (Appendix A). This observation was further confirmed by spatial transcriptomic analysis, which demonstrated elevated GPD1L mRNA levels in HCC tumour regions compared to adjacent normal tissue (Appendix A). Single-cell RNA sequencing analysis also revealed a notable expression of GPD1L within the tumour microenvironment, particularly in natural killer (NK) and T cells (Appendix A).

Interestingly, GPD1L mRNA levels showed an incremental increase, while promoter methylation exhibited a decrease with advancing tumour stage in HCC (Figure 2B,C). Moreover, a strong inverse correlation was observed between GPD1L promoter methylation and gene expression (Figure 2D). There was no significant difference between copy number alterations and GPD1L mRNA levels (Figure 2E). In addition, high GPD1L mRNA expression was associated with TP53 mutant tumours (Appendix A).

Overall, these findings suggest a positive selection for GPD1L overexpression in HCC that correlates with tumour progression. The results also indicate a potential regulatory role of promoter demethylation in driving GPD1L upregulation.

### 2.3. Molecular Characteristics of GPD1L-High Tumours

In order to explore the molecular characteristics associated with GPD1L overexpression, we conducted differential gene expression analysis comparing high and low GPD1L mRNA tumours using the TCGA dataset. This analysis identified 293 up-regulated genes (log2 fold change [FC] > 2, false discovery rate [FDR] *p* < 0.05) and 38 down-regulated genes (log2 FC < −2, FDR *p* < 0.05) (Figure 3). The functional implications of GPD1L expression were investigated by employing gene set enrichment analysis (GSEA) and gene set variation analysis (GSVA). Analysis of Hallmark gene sets revealed that GPD1L-high tumours were enriched for gene sets associated with G2M checkpoint, epithelial-mesenchymal transition (EMT), and E2F target genes (Figure 3 and Appendix A). In contrast, pathways related to bile acid synthesis, fatty acid metabolism, and oxidative phosphorylation were found to be downregulated in GPD1L-high tumours.

To further investigate the biological processes associated with GPD1L, functional enrichment analysis was conducted on genes which exhibited a strong positive correlation with GPD1L expression in HCC tumours. A total of 196 genes were found to have a significant positive correlation (Spearman’s rho > 0.4, FDR *p* < 0.05). Functional enrichment analysis revealed a significant enrichment of E2F-responsive genes (Appendix A and Appendix A), which are known to regulate cell cycle progression, DNA replication, and checkpoint control.

The literature pinpoints miR-210 as a critical regulator of GPD1L in the hypoxia response. However, there was no significant increase in its expression in HCC, and only a weak correlation can be observed between miR-210 and GPD1L expression in the TCGA-LIHC dataset. This suggests that miR-210 regulation of GPD1L may be context-specific and not the primary method of GPD1L overexpression in HCC (Appendix A).

GPD1L’s established function involves converting sn-glycerol 3-phosphate to glycerone phosphate [15]. Examination of the STRING protein–protein interaction network corroborates GPD1L’s position as a hub within a network of highly interacting proteins that regulate glycerophospholipid (GPL) metabolism (Appendix A). Utilising g:Profiler to functionally annotate TCGA-LIHC genes exhibiting strong co-expression with GPD1L, we further confirmed its pronounced correlation with genes linked to phospholipid metabolism, such as LPCAT1 (Spearman’s rho 0.47, FDR *p* value 1.27 × 10^−18^) (Appendix A).

These findings underscore the disruption of E2F-mediated transcriptional programs and metabolic reprogramming associated with GPD1L mRNA overexpression in HCC. When considered alongside the clinical, pathological, and gene enrichment data, we speculate that heightened GPD1L expression is prompted to sustain membrane GPL biosynthesis in rapidly proliferating, aggressive HCC tumours, contributing to an overall lipid metabolic reprogramming.

### 2.4. GPD1L as a Predictive Biomarker for Treatment Response

Finally, we explored the potential of utilising GPD1L as a predictive biomarker for therapeutic response. We analysed the GDSC dataset, which includes FDA-approved drugs potentially not well-studied in HCC, using correlation analysis to identify novel associations with GPD1L expression in HCC cell lines, aiming to uncover unexplored connections and insights into treatment responses. Analysis of GPD1L mRNA levels in 17 hepatocellular carcinoma (HCC) cell lines in the GDSC1 database revealed a robust inverse correlation with therapeutic response, as measured by the half-maximal inhibitory concentration (IC50), for three therapeutic agents including PF-562271, Linsitinib, and BMS-754807 (Figure 4A,B). PF-562271 is an inhibitor that targets focal adhesion kinase (FAK), a key signalling protein involved in cell adhesion, migration, and invasion [16]. Linsitinib and BMS-754807 are inhibitors that target the insulin-like growth factor 1 receptor (IGF1R), a receptor tyrosine kinase involved in cell growth and survival-signalling pathways [17,18].

Additionally, we delved into the potential association between GPD1L expression and the response to tyrosine kinase inhibitors (TKIs), specifically the first-line treatments sorafenib and lenvatinib for advanced HCC. Given that lenvatinib response data are absent from the GDSC dataset, we turned to response data for lenvatinib from the study by Rees et al. [19]. This unveiled a statistically significant inverse correlation between GPD1L mRNA levels in HCC cell lines and the AUC for lenvatinib (Spearman’s rho −0.55, *p* = 0.011), signifying a heightened drug sensitivity that corresponds to elevated GPD1L expression (Appendix A). Conversely, our inquiry indicated a lack of substantial correlation between GPD1L mRNA and sorafenib response, both in the GDSC dataset and the Rees et al. datasets (Appendix A).

To validate the predictive potential of GPD1L in silico, IC50 values were imputed in the TCGA LIHC dataset using oncoPredict. This analysis predicted differential responses to two out of three agents, namely BMS-754807 and PF-562271 (Figure 4C). For in vitro validation, we selected three HCC cell lines with varying levels of GPD1L expression (Appendix A). The cell lines characterised by lower GPD1L expression, namely PLC/PRF/5 and HepG2, exhibited greater resistance to all three drugs compared to the cell line with the highest GPD1L expression (Hep3B). Furthermore, we conducted a knockdown experiment using siRNA to suppress GPD1L expression in Hep3B cells, which resulted in reduced sensitivity specifically to PF-562271, while the sensitivity to Linsitinib and BMS-754807 remained unchanged (Figure 4D,E). These findings indicate the presence of additional regulatory mechanisms beyond GPD1L expression that contribute to the cellular response to IGF1R inhibitors.

These findings indicate that GPD1L holds promise as a candidate surrogate biomarker for response to inhibition of IGF1R and FAK. The observed inverse correlation between GPD1L expression and therapeutic response highlights its potential utility in predicting treatment outcomes and guiding the stratification of targeted therapies in HCC patients based on GPD1L gene expression status, particularly in FAK inhibitors.

## 3. Discussion

The present study investigated the role of GPD1L gene expression as a prognostic biomarker in HCC and explored its potential implications in tumorigenesis, molecular characteristics, and therapeutic response. It is speculated that GPD1L overexpression in HCC may be a consequence of promoter demethylation and the wider E2F dysfunction with advancing tumour stage, where GPD1L itself is an E2F3 target.

Clinical studies have demonstrated that higher GPD1L expression in HCC is associated with an advanced tumour stage, larger tumour size, increased microvascular invasion, and reduced overall and disease-free survival rates [4,5,6,7,8,9,10,20]. In addition to altered gene expression, aberrant promoter methylation levels of GPD1L have been observed in various malignancies [21]. Promoter hypermethylation of GPD1L has been associated with adverse clinicopathological features and reduced patient survival, indicating that promoter-methylation-mediated GPD1L inactivation may contribute to HCC progression [5,6,22]. These findings suggest that GPD1L expression may serve as a prognostic marker for HCC and a potential therapeutic target.

At the molecular level, altered GPD1L expression and activity have been observed in HCC tissues compared to adjacent non-tumour liver tissues. Furthermore, GPD1L plays a role in dysregulated lipid metabolism, a hallmark of HCC, influencing lipid synthesis, lipolysis, and mitochondrial function, which impact tumour cell proliferation, survival, and metastatic potential [23,24]. GPD1L, through its involvement in lipid metabolism and its interaction with AMP-activated protein kinase (AMPK) and mammalian target of rapamycin (mTOR) signalling, plays a critical role in integrating metabolic and signalling cues that contribute to HCC development and progression [6,25]. GPD1L modulates the AMPK pathway, a critical regulator of cellular energy homeostasis, thereby influencing downstream signalling cascades associated with cell growth, metabolism, and apoptosis [6]. Activation of the AMPK pathway can inhibit HCC cell proliferation and induce cell cycle arrest, suggesting that the dysregulation of GPD1L-mediated AMPK signalling may contribute to HCC development and progression [26,27]. Additionally, GPD1L has been shown to affect mTOR signalling, possibly through its impact on lipid metabolism and energy balance [6]. Dysregulated mTOR signalling is frequently observed in HCC and is associated with increased tumour cell proliferation and survival [28].

In addition to its role in HCC, GPD1L has also been implicated in other solid tumours, indicating its broader relevance in cancer biology [7,8,9,10]. For example, in non-small cell lung cancer, GPD1L is involved in the regulation of cell polarity, metabolic reprogramming, and tumorigenesis [8]. These findings suggest that GPD1L may have common functions and regulatory mechanisms across different tumour types, underscoring its potential as a therapeutic target and prognostic marker in various cancers.

In healthy liver tissue, GPD1L primarily serves as a key player in lipid metabolism and energy homeostasis. It participates in glycerolipid biosynthesis, aiding in energy storage and cellular membrane composition. GPD1L’s involvement in the glycerol phosphate shuttle facilitates the transfer of reducing equivalents during glycolysis and oxidative phosphorylation, which is essential for energy production [9,11,15]. In contrast, GPD1L’s functional role in other organs can vary widely and has been linked to processes like cell polarity and adipogenesis [15]. This diversity underscores the tissue-specific nature of GPD1L’s functions, adapting to the distinct metabolic demands and physiological requirements of different organ systems.

A major strength of our study was the use of multiple datasets, including the TCGA-LIHC primary tumour dataset and three additional HCCDB v2.0 datasets, which provided robustness and consistency to our findings. The significant association between high GPD1L mRNA expression and poor clinical outcomes in HCC patients, as well as the independent prognostic significance of GPD1L expression for OS, highlights the potential of GPD1L as a prognostic biomarker in HCC. While our study provided insights into the molecular characteristics associated with GPD1L overexpression in HCC, further functional studies are needed to elucidate the underlying mechanisms. For example, the dysregulation of metabolic pathways observed in GPD1L-high tumours suggests the involvement of metabolic reprogramming in GPD1L-mediated tumorigenesis. Future studies should focus on deciphering the metabolic alterations associated with GPD1L overexpression and evaluating their therapeutic implications.

Furthermore, the need to explore GPD1L’s potential as a surrogate predictive biomarker for treatment response becomes apparent. Our results indicated an inverse correlation between GPD1L mRNA levels and therapeutic response, particularly in relation to inhibition of IGF1R and FAK. The cell line drug testing data should be further validated in patient-derived HCC-tumour organoids characterised by GPD1L expression. Our gene silencing experiments, in fact, underscore the likelihood that the association between GPD1L and IGF1R inhibitor response is not directly functional but rather reflects its interrelation with other influential factors. Conversely, GPD1L’s expression potentially showcases a more immediate mechanistic connection with FAK inhibitor (PF-562271) response. In the future, delving into mechanistic inquiries should encompass the exploration of additional underlying regulatory mechanisms, beyond GPD1L, that potentially contribute to the IGF1R inhibitor response.

In addition, recent research underscores metformin’s immunomodulatory attributes [29], suggesting its synergistic utilisation with IGF1R and FAK inhibitors may enhance therapeutic efficacy in HCC. Leveraging the inhibitory effects of metformin on gluconeogenesis alongside the targeted inhibition of IGF1R and FAK pathways may offer a dual-pronged approach to tackling the intricate molecular landscape of HCC. Further investigation into the intricate interplay among these agents is crucial to optimise their combined potential and elevate the prospects of effective HCC treatment strategies.

Nonetheless, the results obtained from our study have potential implications for clinical practice and personalised medicine in HCC. The identification of GPD1L as a prognostic biomarker could aid in patient stratification and enable more tailored treatment approaches. GPD1L expression status could help guide therapeutic decisions, particularly for targeted therapies such as the inhibition of FAK. The integration of GPD1L assessment into clinical practice has the potential to improve treatment outcomes and patient survival. Prospective clinical trials that stratify HCC patients based on GPD1L expression levels and evaluate treatment response are necessary to validate the predictive potential of GPD1L.

In conclusion, our study provides evidence of the prognostic significance of GPD1L gene expression in HCC and sheds light on its potential implications on tumorigenesis, molecular characteristics, and treatment response. Future work should focus on prospective studies, functional investigations, and clinical trials to fully explore the potential of GPD1L as a prognostic and predictive biomarker in HCC and other solid tumour types. The impact of our results extends beyond HCC and could have broader implications in the field of cancer research and personalised medicine.

## 4. Materials and Methods

### 4.1. Public Databases

Molecular (transcriptomic, copy number alteration, mutation, methylation) and clinical data from the TCGA-LIHC primary tumour dataset (n = 370) were downloaded from the GDC portal using the TCGAbiolinks package in R (accessed on 15 June 2023). Datasets were dichotomised by median GPD1L mRNA value into ‘high’ and ‘low’ GPD1L expression [30,31]. HCC transcriptomic (bulk, spatial, and single-cell RNA-seq) and survival data were evaluated using the online portal at HCCDB v2.0 (http://xsh.mywsat.cn/#/home, accessed on 15 June 2023).

### 4.2. Survival Analysis

Kaplan–Meier and Cox proportional hazards survival analysis was performed using the survival package in R [32]. Survival curves were compared using the log rank test. The following co-variates were included in the multivariate Cox regression analysis after confirming the proportional hazards assumption was maintained: age, sex, resection margin status, stage, ECOG performance status, and tumour purity. Continuous variables (age and tumour purity) were dichotomized by median value. We evaluated overall survival (OS) and progression-free interval (PFI) as recommended by Liu et al. [33].

### 4.3. Differential Gene Analysis

Differential gene analysis was performed in TCGA-LIHC using RNA-seq raw counts data and DEseq2 in R [34]. Genes were ranked by t-statistic and analysed using gene set enrichment analysis software (pre-ranked analysis) and the Hallmark gene set MSigDB v2023.1.Hs (Mar 2023) (Broad Institute, Cambridge, MA, USA, accessed on 16 June 2023) [35,36]. A parallel analysis of the 50 Hallmark gene sets was conducted using gene set variation analysis (gsva) in R to identify differential gene set activity (FDR *p* < 0.05) [37]. Top positively correlating genes (Spearman’s rho > 0.4; FDR *p* < 0.05) were downloaded from cBioportal (https://www.cbioportal.org/, accessed on 20 June 2023) and functional enrichment analysis was performed using g:Profiler (https://biit.cs.ut.ee/gprofiler/gost, accessed on 23 June 2023) [38,39,40]. miRNA analyses were performed and visualised using CancerMIRNome (http://bioinfo.jialab-ucr.org/CancerMIRNome/, accessed on 23 June 2023) [41].

### 4.4. Drug Response Prediction

Drug response (IC 50) (n = 326) and HCC cell line (n = 17) transcriptomic data for gene response correlation analyses were downloaded from the GDSC portal (GDSC1) (https://www.cancerrxgene.org/, accessed on 27 June 2023) [42]. Drug response was imputed in TCGA-LIHC using oncoPredict in R [43].

### 4.5. Cell Culture

The human HCC cell lines Hep3B, HEPG2, PLC/PRF/5 were obtained from the American Type Culture Collection. Cells were cultured in minimum essential medium (MEM, Gibco, Paisley, UK) supplemented with 10% fetal bovine serum (FBS, Sigma, Welwyn Garden City, UK), 1X Glutamax (Gibco, UK), 1X Penicillin-Streptomycin (Sigma, Welwyn Garden City, UK), non-essential amino acids (Sigma, UK), and 1 mM sodium pyruvate (Sigma, UK). Cells were incubated (37 °C, 5% CO_2_, humidified) and passaged using TrypLE Express (Gibco, UK) to maintain 70–80% confluency.

### 4.6. siRNA Transfection

GPD1L siRNA (sc-78210) and the negative control scramble siRNA (sc-37007) were purchased from Santa Cruz (Germany). Hep3B cells were seeded at 80% confluency prior to transfection. For transfection, a 1:1 ratio of siRNA duplex and siRNA transfection reagent (Santa Cruz Biotechnology, Heidelberg, Germany) was diluted in an antibiotic-free siRNA transfection medium (Santa Cruz Biotechnology, Germany) and added to cells. After 6 h incubation, the complete growth medium containing 20% FBS was added into the transfection mixture. After 24 h, the transfection mixture was replaced by fresh media for downstream assays.

### 4.7. Drug Treatment and Cell Viability Assay

Linstinib (OSI-906), PF-562271, and BMS-754807 were purchased from Selleckchem. Cells were seeded at 40% confluency overnight and incubated with drugs dissolved in 0.1% DMSO for 48 h. Viability was measured by the CellTiter 96 AQueous One Solution cell proliferation assay (Promega, Southampton, UK) following manufacturer’s instructions. The optical absorbance was recorded at 490 nm using a BioTek 800 TS microplate reader (Agilent, Cheadle, UK). Data were normalised against the vehicle control.

### 4.8. qPCR

To assess transfection efficiency, total RNA was extracted with RNeasy Kit (Qiagen, Manchester, UK), quantified with Nanodrop One UV-Vis Spectrophotometer (Thermo Scientific, Loughborough, UK), and cDNA was synthesised from mRNA using LunaScript RT SuperMix Kit (New England Biolabs, Hitchin, UK) following manufacturer’s instructions. Luna Universal qPCR master mix (New England Biolabs) was used with the following primers: GPD1L_Fw: 5′-GTTGCCATGTCAAATCTTAGCG-3′; GPD1L_Rv: 5′-GCACTCTCCCAGTGATCTCAT-3′; GAPDH_Fw: 5′-TCAAGGCTGAGAACGGGAAG-3′; GPD1L_Rv: 5′-CGCCCCACTTGATTTTGGAG-3′. The primer sequences were generated using the NCBI BLAST. cDNA was amplified for 40 PCR cycles in technical triplicates with QuantStudio 7 Pro Real-Time PCR System (Applied Biosystems, Warrington, UK) using GAPDH as internal reference gene. Data are presented as expression fold change normalised against vehicle control.

### 4.9. Statistical Analysis

Variables were compared using Welch’s *t* test or Mann U Whitney for continuous variables as appropriate, and Pearson’s chi-square or Fisher exact test for categorical variables. Data have been presented as means with standard deviation (SD) for continuous variables and frequency with percentage for categorical variables. Multiple testing was adjusted using the Benjamini–Hochberg method. All statistical analyses were done using R software version 4.0.2 unless otherwise indicated. Graphing and non-linear curve fitting for transfection and drug dose response experiments were performed with Prism 9.0.2 (GraphPad, San Diego, CA, USA).

## Figures and Tables

**Figure 1 ijms-24-13113-f001:**
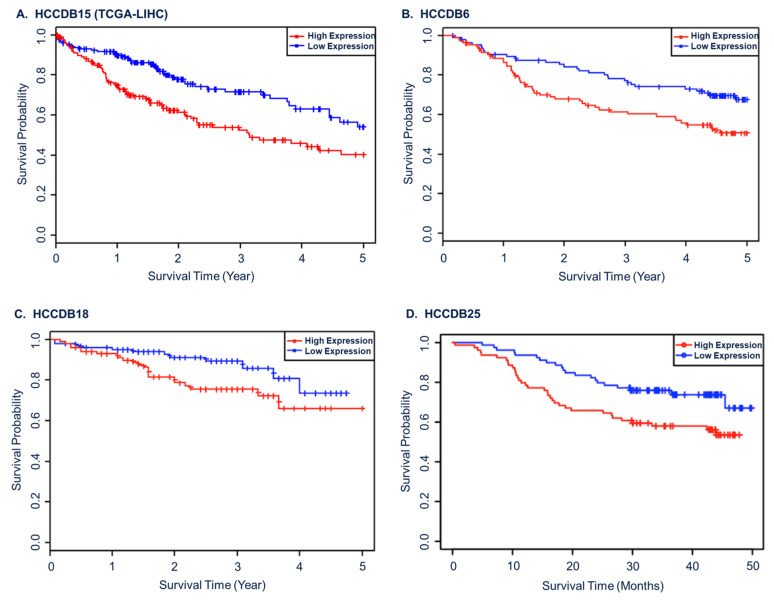
High GPD1L gene expression is prognostic in HCC. Kaplan–Meier plots showing overall survival in HCC cohorts stratified by median GPD1L mRNA level. The datasets included (**A**) TCGA-LIHC; (**B**) HCCDB v2.0 dataset 6; (**C**) HCCDB v2.0 dataset 18; and (**D**) HCCDB v2.0 dataset 25.

**Figure 2 ijms-24-13113-f002:**
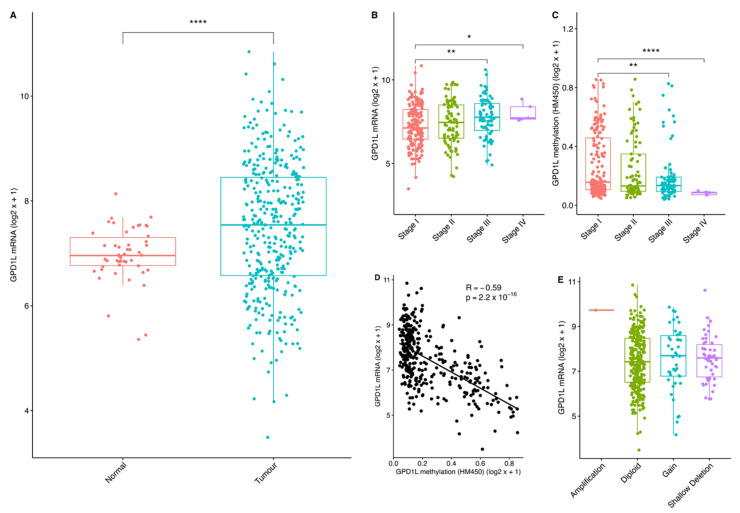
GPD1L overexpression in HCC tumours. (**A**) GPD1L expression in normal (red) and tumour tissue (blue) (TCGA-LIHC); (**B**) GPD1L expression by tumour stage—Stage I (red), Stage II (green), Stage III (blue), Stage IV (purple); (**C**) GPD1L promoter methylation by tumour stage—Stage I (red), Stage II (green), Stage III (blue), Stage IV (purple); (**D**) Correlation between GPD1L methylation and mRNA expression; (**E**) GPD1L expression by copy number aberration—Amplification (red), Diploid (green), Gain (blue), Shallow deletion (purple). Welch’s T test was performed with Benjamini-Hochberg correction for multiple testing. * *p* < 0.05, ** *p* < 0.01, **** *p* < 0.0001.

**Figure 3 ijms-24-13113-f003:**
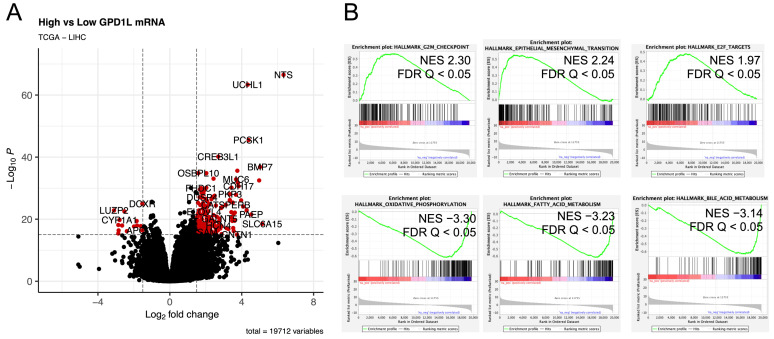
Molecular characteristics of GPD1L-high tumours. (**A**) Volcano plot demonstrating differential genes between GPD1L high and low tumours. (**B**) GSEA enrichment plots for significantly up-regulated and down-regulated Hallmark gene sets.

**Figure 4 ijms-24-13113-f004:**
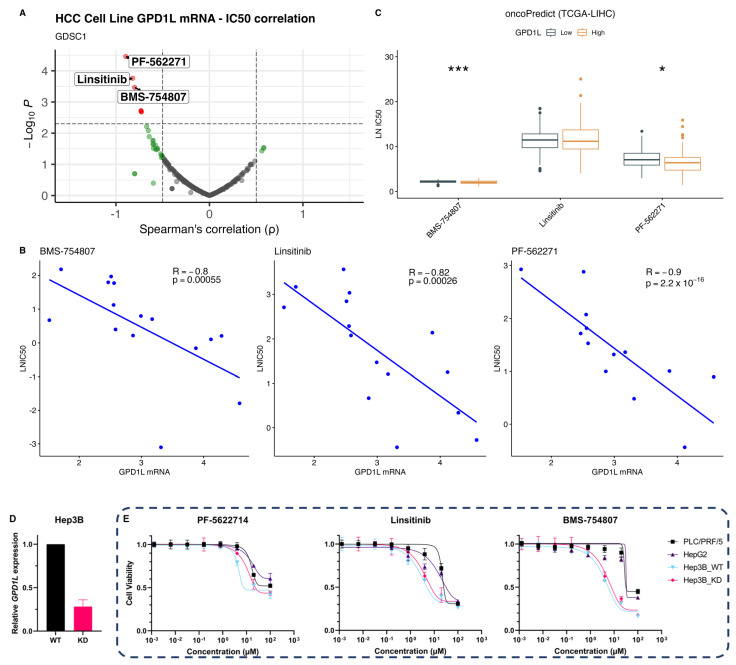
GPD1L as a predictive biomarker for treatment response. (**A**) Volcano plot of correlation between GPD1L in HCC cell lines and IC50 of drugs from GDSC1; (**B**) Top three most significant correlations between GPD1L and drug response; (**C**) Differential analysis of predicted drug sensitivity in TCGA LIHC. Imputed sensitivity generated using oncoPredict and GDSC1 data; (**D**) siRNA knockdown of GPD1L in Hep3B as measured by qPCR (n = 3); (**E**) Dose–response curves of drug treatments with non-linear fit (n = 4). For (**C**), Mann U Whitney testwas performed with Benjamini-Hochberg correction for multiple testing. * *p* < 0.05, *** *p* < 0.001.

**Table 1 ijms-24-13113-t001:** Comparison of GPD1L mRNA expression levels in low and high groups in TCGA-LIHC.

Variable	Overall, N = 370 ^1^	Low, N = 186 ^1^	High, N = 184 ^1^	*p*-Value ^2^
Age (years)	61 (51, 69)	64 (55, 70)	59 (50, 68)	0.003
Sex				0.001
Female	121 (33%)	46 (25%)	75 (41%)	
Male	249 (67%)	140 (75%)	109 (59%)	
Stage				<0.001
Stage I	171 (49%)	103 (57%)	68 (41%)	
Stage II	85 (25%)	46 (26%)	39 (23%)	
Stage III	85 (25%)	31 (17%)	54 (33%)	
Stage IV	5 (1.4%)	0 (0%)	5 (3.0%)	
Resection margin status				0.038
R0	323 (89%)	169 (93%)	154 (85%)	
R1	17 (4.7%)	5 (2.8%)	12 (6.6%)	
R2	1 (0.3%)	0 (0%)	1 (0.5%)	
RX	22 (6.1%)	7 (3.9%)	15 (8.2%)	
ECOG Performance Status				0.004
0	162 (57%)	96 (62%)	66 (50%)	
1	84 (29%)	46 (30%)	38 (29%)	
2	26 (9.1%)	11 (7.1%)	15 (11%)	
3	12 (4.2%)	1 (0.6%)	11 (8.4%)	
4	2 (0.7%)	1 (0.6%)	1 (0.8%)	
HCC primary risk factor				0.20
Alcohol consumption	117 (33%)	63 (36%)	54 (31%)	
Alpha-1 antitrypsin deficiency	1 (0.3%)	0 (0%)	1 (0.6%)	
Hemochromatosis	5 (1.4%)	2 (1.1%)	3 (1.7%)	
Hepatitis b	80 (23%)	40 (23%)	40 (23%)	
Hepatitis c	34 (9.7%)	23 (13%)	11 (6.3%)	
No history of primary risk factors	91 (26%)	37 (21%)	54 (31%)	
Non-alcoholic fatty liver disease	12 (3.4%)	6 (3.4%)	6 (3.4%)	
Other	11 (3.1%)	6 (3.4%)	5 (2.9%)	
INR	1.1 (1.0, 9.1)	1.1 (1.0, 8.9)	1.1 (1.0, 9.5)	0.60

^1^ Median (IQR); n (%); ^2^ Wilcoxon rank sum test; Pearson’s Chi-squared test; Fisher’s exact test.

## Data Availability

The data presented in this study are available within the article or Appendix A. The results here are in part based upon data generated by the TCGA Research Network: https://www.cancer.gov/tcga (accessed on 15 June 2023).

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
