# Peer review of "Prognostic and Predictive Utility of GPD1L in Human Hepatocellular Carcinoma"

_ijms, 2023, doi:10.3390/ijms241713113_

Round 1
Reviewer 1 Report
The study analyzed independent datasets and found that high GPD1L expression was significantly correlated with poor survival in HCC patients. GPD1L expression was elevated in tumor tissue compared to adjacent normal tissue. As the tumor stage advanced, GPD1L expression and promoter demethylation increased, suggesting positive selection during tumorigenesis. GPD1L overexpression was associated with dysregulated metabolic pathways and enrichment of gene sets related to cell cycle control and epithelial-mesenchymal transition. Additionally, an inverse correlation was observed between GPD1L expression and response to certain therapeutic agents (PF-562271, Linsitinib and BMS-754807), indicating its potential as a predictive biomarker for HCC treatment outcomes.
Overall, GPD1L expression in HCC appears to have prognostic significance and is associated with specific molecular characteristics. It may serve as a potential therapeutic target and predictive biomarker for treatment response, particularly in relation to inhibition of IGF1R and FAK.
Comments:
1) I recommend delineating the criteria for testing a reversible inhibitor of FAK, especially when there are no existing clinical trials for its usage in the context of HCC. Providing clear criteria for testing the inhibitor will help readers understand the rationale behind its potential therapeutic application and contribute to a more comprehensive understanding of its potential benefits and risks in HCC treatment.
2) There is no specific mention of the efficacy of GPD1L as a biomarker in the context of sorafenib and lenvatinib treatment for HCC. The article focused on the prognostic significance, molecular characteristics, and potential predictive biomarker role of GPD1L in HCC, but it did not include data related to its specific efficacy as a biomarker for sorafenib and lenvatinib response.
3) Metformin's multifaceted mechanism of action involves inhibiting gluconeogenesis through various intracellular pathways, including AMPK activation and cAMP reduction, resulting in its glucose-lowering effects. A recent article (PMID: 37370771) suggests that utilizing metformin's immunomodulatory properties in combination with other therapeutic strategies could provide a hopeful avenue for enhancing treatment options for hepatocellular carcinoma in the future. I strongly suggest, that these could be briefly discussed in the Discussion section.
Overall, the English language in the article is good, with only minor room for improvement.
Author Response
Document attached

Reviewer 2 Report
The present study analyzed the expression profile of Glycerol-3-phosphate dehydrogenase 1-like (GPD1L) in hepatocellular carcinoma (HCC) and its relation to the prognosis of HCC patients. The authors showed the high expression of GPD1L is related to the poor prognosis of HCC patients. Analysis of Hallmark gene sets revealed that GPD1L-high tumors were enriched for gene sets associated with G2M checkpoint, epithelial-mesenchymal transition (EMT), and E2F target genes. The expression level of GPD1L was related to the drug sensitivity of PF-562271. Although the reviewer considers the present study contains the novel results and the manuscript is well-written, it is difficult to accept the present manuscript. The reviewer would like to describe some comments as described below.
1. The authors showed the relation between the high GPD1L expression and the poor prognosis of HCC patients. As described in the present manuscript, this result is not novel. The reviewer would request to investigate more detailed relationship between the GPD1L expression and worse tumor behavior. Is there any relation to the clinicopathological factors, e.g. the tumor differentiation, the vascular invasion and the intrahepatic metastasis?
2. The authors showed the potential of GPD1L as a predictive biomarker for treatment response to the inhibitor of IGF1R or FAK. The reviewer remains in doubt regarding the effectiveness of GPD1L as a predictive biomarker for treatment response. The reviewer understood the difference of IC50 in the HCC cells with different GPD1L expression levels. But the reviewer considers that the IC50 is not largely changed by the knockdown of GPD1L expression particularly in IGF1R inhibitors. Therefore, the other factors, not GPD1L, are suggested to cause the difference of IC50 between the cells with low GPD1L expression (PLC/PRF/5 and HepG2) and those with high GPD1L expression (Hep3B).
3. The authors showed the expression of GPD1L is associated to the various biological events according to the results of several in-silico analyses. The reviewer would request the authors to indicate which biological events have more significant relation to the expression of GPD1L. If it can be indicated, the functional importance of GPD1L in HCC can be considered more deeply in the present manuscript.
4. The reviewer can not understand the method of separating the HCC patients into two groups (Low and High GPD1L). The reviewer would request the criteria of the expression level of GPD1L in separating the HCC patients.
5. According to the article published by Nagase T et al. (Nagase T et al., DNA Res. 2 (1): 37–43, 1995.), GPD1L (described as KIAA0089 in this article) is ubiquitously expressed in various tissues but not in liver tissue. The reviewer considers that the functional role of GPD1L is different in liver tissue from the other tissues. The reviewer would request the authors to consider the expression of GPD1L in normal liver tissue and describe some comments in the discussion of the present manuscript.
The reviewer considers the quality of English is sufficient.
Author Response
Document attached

Round 2
Reviewer 2 Report
The reviewer has been satisfied with the authors' comments and revisions.
The reviewer considers the quality of English is sufficient.